# The Emergence of Lotus Farming as an Innovation for Adapting to Climate Change in the Upper Vietnamese Mekong Delta

Hoang Thi Minh Vo [1,2,]*, Gerardo van Halsema [1], Petra Hellegers [1], Andrew Wyatt [3] and Quan Hong Nguyen [4,5]

1  Water Resources Management Group, Wageningen University & Research (WUR),
   6708 PB Wageningen, The Netherlands; gerardo.vanhalsema@wur.nl (G.v.H.); petra.hellegers@wur.nl (P.H.)
2  Faculty of Environment, University of Science, Vietnam National University Ho Chi Minh City (VNU-HCM),
   Ho Chi Minh City 70000, Vietnam
3  International Union for Conservation of Nature (IUCN), Ho Chi Minh City 70000, Vietnam;
   Andrew.WYATT@iucn.org
4  Institute for Circular Economy Development (ICED), Vietnam National University Ho Chi Minh
   City (VNU-HCM), Ho Chi Minh City 70000, Vietnam; nh.quan@iced.org.vn
5  Center of Water Management and Climate Change (WACC)—Institute for Environment and Resources,
   Vietnam National University Ho Chi Minh City (VNU-HCM), Ho Chi Minh City 70000, Vietnam
*  Correspondence: vtmhoang@hcmus.edu.vn; Tel.: +31-630428399

**Abstract:** Climate change poses an acute threat to the Vietnamese Mekong Delta (VMD). To respond to this threat, the structure of the delta's agriculture-based economy must transform, becoming more adaptive to changing conditions. One adaptive livelihood option is the use of flood-based farming systems in the upper VMD. The present study examines local perceptions of such a system in Dong Thap Province, a lowland, flood-affected area of the upper VMD. Specifically, we explored lotus farming as a potential flood-based adaptive livelihood model for the region. The study advances the current literature by using historical research, embedded in narrative analysis applying the social construction of technology (SCOT) lens. We collected data through in-depth interviews and field surveys to qualitatively analyze the emergence and status of lotus cultivation in the study area, zooming in on how local society evolved with expansion of this farming model. The findings suggest that as an innovative idea, lotus farming initially emerged due to unfavorable natural conditions, and then was developed as an attractive nature-based livelihood, and thus received increased attention. It has been accepted and modified over time according to the new interests of further stepped-in stakeholders. Our findings echo the social construction of technology perspective as we found lotus farming to be a technological artifact that did not develop on its own, by was driven by different interpretations and re-negotiation process. This made more room for stakeholders to shape and reshape it in a way that fits their interests.

**Keywords:** lotus farming; innovation; historical approach; stabilization; Vietnamese Mekong Delta

## 1. Introduction

Climate change poses an acute threat to the Vietnamese Mekong Delta (VMD). As the region's economy is predominantly agriculture-based [1], the Vietnamese government today faces a dilemma of whether to continue pursuing the traditional model of farming or to explore farming systems better adapted to the changing climate. Many scholars have studied innovative ideas in flood management (including flood-adapted livelihoods), but the uptake of novel approaches has been slow, partly due to the nature of Vietnam's existing planning system, which is top-down and still fragmented [2–4]. Several new strategic plans have been developed and introduced, including the 2013 Mekong Delta Plan (MDP), which proposes major changes in the development pathway of the VMD. One of the main models explored in the MDP is flood-based farming (FBF), as an alternative to the intensive

rice monoculture covering much of the upper VMD [5,6]. Furthermore, the MDP calls for planning processes to involve a broad spectrum of stakeholders, with grassroots actors playing a pivotal role in agricultural innovation. Grassroots actors, particularly farmers, are key due to their role not only as "receivers" of innovations but also as producers and holders of knowledge related to innovations and innovation creation [7]. Other relevant social groups (RSG) in Vietnam are government, academia, and donors. All of these stakeholders have played a role, whether active or passive, in shaping innovations in the country. Their varied interests and agendas, depending on how consensus was reached, have thus influenced the outcomes of planning processes. Recent studies by our project team (UDW—Urbanizing deltas of the word) point to a shift in mindset among Vietnamese political actors (an important stakeholder) toward recognition of the need for adaptation pathways to ensure a more sustainable future [8–10].

FBF has been proposed as a particularly promising "innovation" for sustainable agriculture in the upper VMD, as it could help farmers harness the changing climatic conditions for the goals of food security and sustainable livelihoods [11]. Among others, lotus cultivation has emerged as a technology that fits in the FBF frame. Lotus cultivation can be combined with additional services, such as aquaculture, water retention, and tourism, bringing enhanced livelihood options. Lotus has certain advantages compared to rice cultivation. First, it requires less investment in fertilizers and pesticides. Second, it suits the land and water conditions of FBF in the upper VMD very well. Whereas intensive rice production requires floodwaters to be minimized, lotus can adapt to either high or low water levels. As floodwaters can be allowed to enter lotus fields, thus enriching soils and creating habitats for wild fish and other aquatic species, lotus cultivation can be part of a system of floodwater storage for dry season usage [6].

The MDP calls for innovations appropriate to the changing natural conditions of the delta, and lotus farming may fit the bill as a suitable adaptation for floodplain areas. The current study investigates the lotus cultivation model (including intensive lotus, lotus-rice, lotus-fish, and lotus-ecotourism) as an appropriate innovation for the VMD floodplains. Though lotus is a long-existing farming practice pre-dating the MDP, the planning strategy frames lotus cultivation as a suitable, climate-adaptive technology that provides good multifunctionality. These are considered basic requirements for a livelihood option to thrive on the natural floodplains of the Plain of Reeds. Lotus also provides opportunities for diverse associated products. Indeed, every part of the lotus plant can be harvested, processed, and marketed for different uses.

The current study opens a discourse on how lotus farming as a technology has been designed, implemented, adopted, and spread. We, therefore, adopted historical research as our main method, using the social construction of technology (SCOT) lens. SCOT views technologies as a result of stakeholder interaction, with human actions, understood as shaping technology, rather than technology driving human action [12]. SCOT has commonly been used to study state-of-the-art technologies and governance innovations [13–20], globally. In Vietnam's existing literature body specifically, there are theoretical gaps concerning SCOT, and empirical gaps as it is unclear how innovation has been perceived, interpreted, and accepted among relevant stakeholders. This paper attempts to bridge these gaps. This paper applies the historical research method embedded in narrative analysis using the SCOT lens to trace the emergence of lotus farming as an alternative to rice in the upper VMD. In addition to providing a historical overview of the developments and changes affecting lotus farming in the VMD, we address three research questions: (a) Why is there so much flux in the lotus farming model as an innovation? (b) What features of lotus farming are working well? (c) Has lotus farming reached some degree of closure or stabilization that could confirm its suitability as a climate-adaptive innovation for the VMD; or is the search for an appropriate lotus technology still underway?

## 2. Materials and Methods

### 2.1. Historical Research Method

Historical research in the social sciences is a method for discovering, from records and narrative accounts, what happened in some past period [21]. This method enables the researcher to describe important and interesting past events by listing or drawing them in chronological sequence. Historical research is at once descriptive, factual, and fluid, not merely nostalgia. The method "attempts to systematically recapture the complex nuances, the people, meanings, events and even ideas of the past that have influenced and shaped the present" [21] (p. 115). Data for this method can be gained from primary sources that provide descriptions and witnesses to the focus events and time. For instance, one such primary source is individuals who are still alive and can bear witness to some aspects of the research focus. The current study employed this means of data collection [21], embedded in the narrative analysis [22].

Our approach was grounded in and reflects the SCOT lens. The central premise of the SCOT perspective is that technology is not an isolated artifact developed by engineers in the laboratory and then implemented in the field. Rather, it is an outcome of social interactions between scientists, developers, and users [23]; a product of human choices and social processes [24]. Technological innovation is a continuum from inception to development, testing, application, and ongoing modifications and iterations [12]. Intended users and effects may not materialize, while other uses or effects may become apparent, and users may alter the technology for unforeseen purposes and uses. In analyzing lotus farming as an innovation, we viewed it not just as a technology (a knowledge-transferred term) to the delta planning management, especially for the VMD, but as an "actant" (the terminology used by [23]) or "artifact" (that is shaped by social actors [23]) in the delta.

### 2.2. Research Design, Case Study, and Data Collection

To investigate the development of rotational floating lotus and how this innovation has been changed and adopted by society, we used a qualitative, explorative research design. We started with a literature review, which fed into open-ended interviews conducted in line with a pre-developed interview guide.

We selected Dong Thap Province as our case study area for several reasons. First, it is located in the upper VMD, which is one of Vietnam's most vulnerable regions [25,26]. Second, the region has attracted the attention of development partners such as the International Union for Conservation of Nature (IUCN) because of local farmers' knowledge of flood-based livelihoods, such as lotus farming. Additionally, the World Wildlife Fund (WWF), seeking to promote sustainable flood-based livelihoods, entered the scene with funding from the Hongkong and Shanghai Banking Corporation (HSBC) to develop lotus-ecotourism in farming areas around Tram Chim, a township in Dong Thap Province. For decades, local farmers have practiced flood-based lotus farming here, with apparently satisfactory results [5].

Applying snowball sampling [27], we selected interview subjects in the research area. The first subject (a farmer) was fixed a priori. We began by interviewing farmers who had been involved in the development of lotus farming. We asked them to identify relevant social actors, which led to our identification of five relevant social groups (RSGs): (i) farmers who had a direct link to the technology and played a part in its development; (ii) scientists, who played a key role in technology design and development; (iii) local officials, including government authorities and farmer representatives, who were indirectly linked to the technology and involved as advocates in policymaking, lobbying, and the process of technology design and development; (iv) development partners, who also acted as advocates and had close links to local officials, while playing the role of both implementer and financial supporter; and (v) traders, including individuals and businesses, who were indirectly linked to the technology and made choices about, bought, and used the technology, while also supplying needed inputs to farmers.

Respondents were selected using stratified purposive sampling [28]. In particular, we sought out farmers who had been involved in or planned to implement lotus farming. We were interested in both the common and not-so-common experiences of farmers. We also reached out to local officials to gain a more comprehensive picture. To make the sample empirically meaningful and accommodate nuances and variations, we sought heterogeneity in terms of social characteristics and experiences [29]. In total, we conducted 31 in-depth interviews. One interview subject was from IUCN, four were local officials, and three were traders. The remaining 23 were farmers. Some of these farmers had been involved in a lotus farming project conducted by IUCN Vietnam, which provided some financial support for the lotus farming operation. Most interviewees were men and the main breadwinner of their household. Ages ranged from 21 to 65 years.

During our open-ended, guided interviews, respondents were encouraged to freely share their experiences and recount stories to illustrate their perceptions. Interviewees could elaborate on what was important to them, while the interview guide kept the discussions on track and ensured that each interview covered specific themes. There was also probing and cross-checking based on earlier answers, and interrogation of information provided by other respondents (without disclosing their identities). The Appendix A presents the interview guide. The interviews were recorded, transcribed verbatim, and thematically coded [29].

## 3. Results

### 3.1. The Story of Lotus Farming and Its Development Stages with Related Technical Issues

Every technology has a legal dimension, it has a history, it entails a set of social relationships, and it has a meaning [24]. Framed by this description, we picked out different pieces of historical stories, told by our 31 interviewees, to bring together the story of floating lotus as an emergent innovation in the upper VMD (Figure 1). Including the historical aspect in the narrative brought out key dynamics that shaped rural change processes in the region [30]. The narrative presented below was constructed from the recollections of our interview respondents. Though largely consistent, our reliance on respondents' recollections does present some intrinsic limitations (e.g., one-side observations with lack of a fuller view on facts and events). It would be preferable to confirm the story constructed through rigorous scholarly analysis.

#### 3.1.1. Land Reclamation (1975)

The story of lotus farming began following the war in Vietnam when the Plain of Reeds was still covered by forest and the soils were classified as mainly acid sulfate. Local people call this period "Exploring the New Economic Zones". Each household received 10 ha of uncultivated land, which led to the clearing of forests for agriculture, mostly rice cultivation. However, rice yields were very low, due to the unsuitability of the region's agroecology for this crop. This period of land reclamation was important in setting the stage for the emergence of lotus, as farmers found that they could hardly generate an income with rice cultivation. At that time, lotus was chosen to be cultivated in the region. Initially, farmers planted lotus mostly by using seedlings. The seeds were brought in from other places, then propagated for continued use over many years. For harvesting, farmers collected lotus seeds and sold these spontaneously to individual traders. Lotus yields were reported to be very low.

#### 3.1.2. Taiwanese Export Attempt and Intensive Lotus (1980)

The Taiwanese pioneered commercial lotus farming in the VMD. Cultivation of lotus was believed to flush out acid sulfate and "clean" and enrich the soil. In 1980, the Cao Tung Company arrived from Taiwan and leased huge areas of acid sulfate lands for lotus farming, starting with intensive lotus cultivation to produce lotus seed for export. This seed was in high demand in the Taiwanese market. The Cao Tung Company invited local farmers to join the lotus campaign, initially providing seeds, fertilizers, cultivation techniques, and

knowledge. The Taiwanese helped the farmers and offered a guaranteed output market for lotus products ranging from flowers to the rhizome, leaves, leaf stalk, stems, and seed. However, they set high standards for these products. This was the first difficulty confronted by lotus development in the region. Local farmers could not conform to the high standards, so the Cao Tung Company departed after two lotus crops. Nonetheless, still subscribing to the idea brought by the Taiwanese firm, local farmers continued to grow lotus, though for an internally oriented market rather than the international (particularly Taiwanese) market the Cao Tung Company had targeted. As farmers at that time could no longer produce for export to Taiwan, some form of local trade must have emerged (i.e., retailers, supermarkets, or open markets).

### 3.1.3. Diversification of Lotus Farming Types (1994)

Some local farmers were unable to join the tourism model, for example, because they did not own big enough areas of land or did not have land near main roads. They explored livelihoods involving intensive lotus or combinations of lotus with rice, fish, and ecotourism. Each of the lotus models required certain water management techniques. For example, rice-lotus had to be practiced outside dike compartments and needed to adhere to a particular seasonal calendar to achieve high yields. Lotus-fish required a strong high dike, so farmers had to switch from a natural flood regime to controlled flooding. As noted, lotus-ecotourism required the availability of a large land area with adjacency to roads. All types of lotus farming required a suitable flood-control infrastructure and water management capability. For intensive lotus, the availability of high-quality seeds was important.

In parallel, the lotus market expanded, and demand became more diversified. Product innovations also got underway, with items such as lotus wine, tea, medicine, food, and (organic) wrapping materials produced. In the extremely competitive market that opened up, the largest share of profit accrued to traders. The number of lotus traders thus increased, though most operated as individual tradespeople. Interviewees further mentioned "lotus collectors, retailers, wholesalers, supermarkets and open markets" as trade outlets. The biggest market, according to our interviewees, was supermarkets, most of which were located in a few big cities, such as Ho Chi Minh City. Many lotus processing factories were established there and in the neighboring industrial province of Binh Duong. Lotus raw products were typically transported and processed in these urban centers to prepare them for their final markets. Though markets emerged for all parts of the lotus plant, farmers needed specialized knowledge and technical skills to properly harvest the various components. Some farmers did limited "on-site" processing, for instance, peeling off lotus seed skin. This job could be done by hand, without the need for machines. Peeled lotus seed fetched a higher market price than unpeeled seed. This activity reflects a skill that farmers' picked up from their experience with lotus; those who applied it were able to improve their income. Yet, labor shortages undermined the success of lotus farming, as lotus farming is heavy work. Lotus cultivation had always been laborious and became particularly so when farmers discovered a new lotus disease. It had not yet been resolved at the time of this writing. The quotes below from our interviews illustrate the situation:

> "We don't know what happened to our lotus trees. They were all sick ... the lotus rhizome was suddenly rotten.
>
> Scientists and professors from CTU [Can Tho University] already came and took samples, they are investigating. We may know the results soon.
>
> We tried many traditional ways to solve it, but the disease is still there."

### 3.1.4. Lotus-Ecotourism

A group of Vietnamese students visiting the area by chance was amazed at the beauty of lotus and came up with the idea of lotus-ecotourism. This would combine intensive lotus cultivation with fish farming and tourism activities. The idea immediately took off and was developed into a tourism business by Sachi, a Vietnamese company based in

Ho Chi Minh City. Sachi's involvement brought opportunities for good jobs and better incomes for local farmers. At first, Sachi's model worked well, bringing positive change to the region. Many tourists visited the area, attracted by the beauty of lotus, the surrounding landscape, and the local cuisine. Unfortunately, the productive cooperation quickly came to an end, due to a conflict that arose between the investor and the farmers. The problem was that local farmers, who owned the lands, wanted a larger share of the profits from both the lotus crop and ecotourism than the investors were willing to provide. As a result, Sachi left the area. Again, local farmers continued lotus farming, expanding its scale. One by one, many established their own lotus-ecotourism businesses. A weakness, however, was the lack of coordination among the farmers. The lotus-ecotourism initiatives were fragmented, resulting in the replacement of the larger lotus expanses with small-scale cultivation, which detracted from its previous beauty.

This was, however, the first transition in the region from agriculture-based livelihoods to business-based livelihoods. Those who shifted to lotus-ecotourism were able to change their life significantly. They moved up a rung in society. Thanks to lotus-ecotourism, owners began to earn substantial incomes and continued to do so up to the time of our research. They seemed to face very little risk of failure as well, enjoying relatively steady earnings. This reflected a shift in lotus farming, as its function and operators became more diverse. The meaning of lotus farming for grassroots stakeholders changed accordingly, as lotus was now no longer just an agricultural product, but also a landscape element that attracted tourism. This change had far-reaching positive effects on local society. However, the fragmentation of individual landholdings remained a major technical issue hindering the development of lotus-ecotourism in the period. Few farmers could meet a key requirement of the lotus-ecotourism model: maintenance of a beautiful landscape. Some of the farmers interviewed expressed a desire to buy lands from their neighbors, in order to make the larger connected landscapes that tourists paid to see. The fragmentation of landholdings also presented water management issues.

### 3.1.5. Official Support (2014)

Local authorities supported the emergence of the lotus economy. They sought to facilitate links between lotus farmers and other channels and even gave lotus farming households seed and start-up funding to encourage them to plant lotus. This was in accordance with a national policy to promote a restructuring of the agricultural sector toward high-value crops [31]. In Dong Thap Province, lotus farming remained an established practice, though the lotus farming model continued to be modified for improved efficiency and economy. Multiple RSGs entered the scene, promoting lotus farming and searching for better ways to develop lotus. Nonetheless, due to the various constraints and problems that arose, the success of the farming model remained uncertain.

### 3.2. *Types of Lotus Farming and the Technological Development Involved*
### 3.2.1. Intensive Lotus

Intensive lotus can be farmed either inside or outside dike compartments. Among our surveyed farmers, 19 households practiced intensive lotus outside dike compartments and one inside. Among the 19 farms, 79% applied the tilling method (using existing lotus roots), 16% used the method of transplanting lotus rhizomes, and 5% planted new seedlings. The first method was preferred, as farmers believed it helped remove residual matter left from previous crops, thus reducing the risk of plant disease. The second method involved a single transplant operation, in which farmers planted the lotus rhizome at the proper depth, after tilling or cutting down the stalks left from the previous crop. In choosing seeds for intensive lotus, farmers tended to pick Taiwanese varieties, as these produced plants meeting this market's high standards (including lotus buds, seed, and the rhizome). All of the households that used Taiwanese varieties agreed that these were well suited to the local alkaline soils and flood conditions, while also being resistant to pests and diseases and well-adapted to the local climate. Harvesting times depended on the planting method

used. Seedling plantings required 100–120 days to harvest, while rhizome plantings needed 75–90 days.

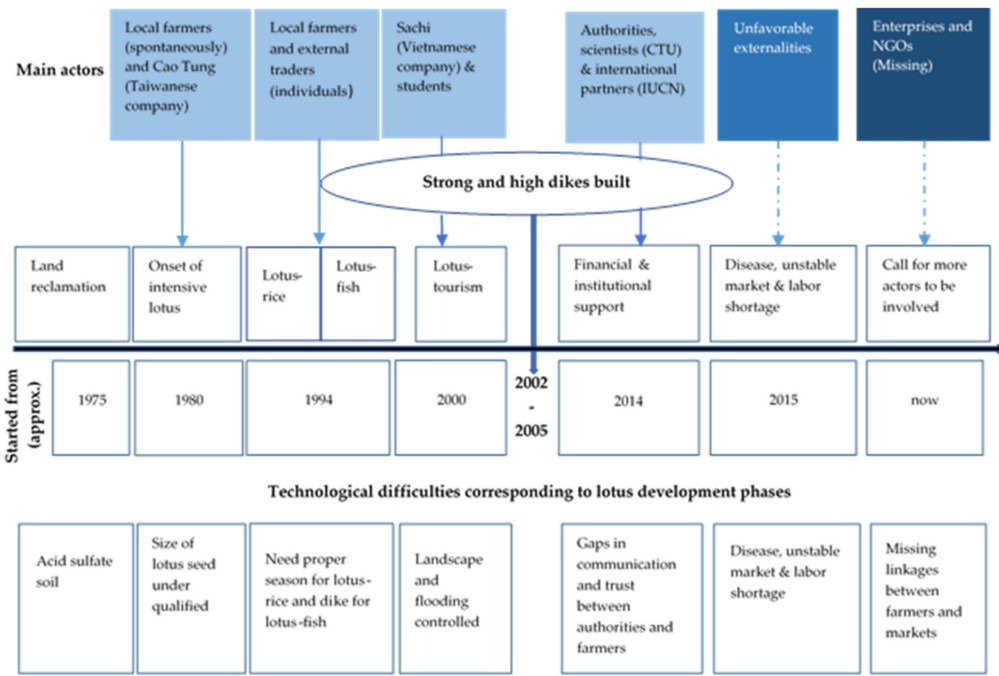

**Figure 1.** Lotus Cultivation: Timeline of Technology Emergence and Involvement of Relevant Stakeholders.

### 3.2.2. Lotus-Fish

Only one household in the research area practiced combined lotus-fish. In that regard, this farmer was a pioneer in the region. The lotus-fish model required investment in a high and solid dike, and the area available for lotus cultivation was reduced so as to increase the water surface area available for fish farming. For this, the ditches around the banks of the lotus field were used. Fish were fed on the food naturally provided by the lotus plants. The farmer, therefore, did not need to purchase supplementary feed for the fish. Caring for the fish was also not technically difficult. Compared to the intensive lotus model outside the dike ring, combined lotus-fish required a higher initial investment to dig ponds and reinforce the ring dike. Therefore, the lotus-fish combination needed to return greater earnings to be an attractive alternative.

### 3.2.3. Lotus-Rice

The integrated lotus-rice model could be practiced outside the dike areas on lands used for double rice. Because the soils in the study area were more suitable for lotus cultivation than for rice, farmers adopted lotus as an alternative to rice and were rewarded with higher incomes. Lotus and rice were cultivated in turns. Rice was planted in the winter-spring season when the profit from the rice was normally high. In the autumn-winter or summer-autumn seasons, lotus was cultivated. The lotus-rice model did require the purchase of pesticides and fertilizers for the rice crop; and some fertilizers were needed for lotus as well, to prevent disease and enrich the soil.

### 3.2.4. Lotus-Ecotourism

The lotus-ecotourism model was implemented on lands with accessibility to roads. In addition to a lotus field, lotus-ecotourism required ponds in between for wild fish and other aquatic species. Farmers practicing lotus-ecotourism also had to reserve areas of land for recreational purposes and restaurants. Materials for these services were mostly "green" such as Melaleuca wood, bamboo, and Nipa palm. Lotus products were served as food in restaurants; in particular, lotus seed was used in various dishes. Leaves were used

for wrapping, and lotus wine, milk, and tea were on offer. Profits from this model arose from intensive lotus cultivation in the lotus fields and the wild aquatic products, alongside supplemental income from tourism businesses and restaurants.

### 3.3. Stakeholder Perceptions of Lotus Farming during Technology Emergence

### 3.3.1. Local Farmers in the Land Reclamation Period

As seen from the story presented above, local farmers early on recognized the value of the lotus plant. They identified the lotus plant as a workable natural crop that could grow under extremely poor natural conditions, and could even actively help to clean and reclaim the soils for agriculture. Lotus thus provided both a means and a purpose for reclamation in extremely poor growing conditions.

### 3.3.2. The Taiwanese Company

The Taiwanese company's idea to commercially develop lotus production in the study area represents a point of stabilization of the lotus farming idea. As farmers came into contact with the high-quality, high-value export-oriented niche market for lotus products, they saw an opportunity to expand their lotus production area. The company provided agronomic and technical support to expand its production base. However, farmers had difficulty meeting the quality and product standards of the Taiwanese side. This led to a scenario in which the market developer and innovator abandoned the scene. Yet, lotus cultivation remained. Products and processing continued, though redirected to the domestic market.

### 3.3.3. Actors in the Emerging Value Chain

Both domestic and international lotus markets became established, though farmers' links with these markets remained lacking. The emerging value chain was shown by the high prices paid for lotus products in 2014–2015 (spring season), which peaked at a level seven times higher than necessary to earn a profit. Interviewees benefited from this development. Whereas they could clear a profit if lotus seed was sold for an average price of US $0.43 per kilo, peak prices ranged from US $2.17 to US $3.13 per kilo. Thus, the market for lotus and lotus products appeared to be large, though farmers in the study area were largely unaware of where the lotus products went.

Expansion of the lotus market was also reported in other Mekong Delta provinces, including An Giang, Long An, Soc Trang, Can Tho, and Vinh Long. Dong Thap Province had more than 850 ha of lotus, while the biggest market was in Ho Chi Minh City, where many large supermarkets were located. In terms of the marketing of lotus at the research site, all lotus trade was conducted between the farmers (lotus growers) and individual traders. These traders then sold the products to bigger companies, retailers (such as supermarkets), wholesalers, or at open markets. In doing so they availed of their own personal connections to these partners. Some companies did attempt to collect lotus products directly from farmers in Dong Thap Province. One of these companies, Nam Huy Trading, Ltd., located in Dong Thap, offered a guaranteed market for some lotus products. In the spring of 2016, it exported 8 tons of lotus products to Korea and Singapore from Dong Thap. Ramsa Corp, in collaboration with Can Tho University (CTU), was developing lotus milk, to be sold under the "zen milk" label for the health food market. The Plain of Reeds Investment and Trade Joint Stock company developed a lotus wine, labeled "pink lotus wine", which many tourists bought as a gift. However, these locally active companies were still very few. While most did have factories in the research area to process raw lotus products, some were still transporting raw products elsewhere for processing before being traded. Some interviewees suggested that the establishment of a robust local-level lotus processing chain could provide better income potential for local lotus farmers.

### 3.3.4. The Vietnamese Company

Sachi, located in Ho Chi Minh City, entered the scene attracted by the great beauty of lotus. Lotus was at that time perceived and promoted as a characteristic natural landscape, in stark contrast to the monoculture rice fields. Sachi's involvement coincided with the economic rise and greater prosperity of Vietnam, especially in the capital city, where a new middle class was emerging, alongside domestic and foreign tourism and leisure industry. Lotus-ecotourism was well marketed and brought a profitable diversification opportunity for lotus farmers. However, this scenario quickly came to an end when a conflict developed over profit-sharing, prompting the key innovator to leave the scene. Lotus combined with tourism remained, but in a fragmented landscape. This fragmentation threatened to undermine the beautiful scenery, which was what brought tourists to the area.

### 3.3.5. Regional Authorities

Advocacy of lotus farming was observed in the involvement of local government authorities, which began to support lotus cultivation and even provided limited financial assistance to lotus farmers. Their intervention was also stimulated by a national policy to restructure agriculture toward high-value crops. Authorities' support for lotus stemmed too from the symbolic value lotus acquired, from the mid-1990s up to the present day, as an embodiment of the region's cultural and landscape identity. This appreciation extended from local residents to government authorities and entered the national conscience. For example, references to lotus abound in Vietnamese literature. Dong Thap Province was promoted as "the lotus region of the delta", with traditional, nature-based landscapes that embraced the natural conditions of the Plain of Reeds and provided rich opportunities for biodiversity preservation and livelihoods for its farmers and residents. Lotus products were thus firmly linked with nature values and a landscape worthy of domestic and foreign appreciation. Here, again, however, some missing links can be identified, such as how to maintain the beauty of the landscape in the context of fragmented farm operations, and how to attract local processing and industry around lotus-based products.

### 3.3.6. Scientists and Development Partners

At the later stages of technology emergence, Can Tho University (CTU) and IUCN (funded by the Coca-Cola Foundation) became involved. In recognition of the need for climate change adaptation, and acknowledging the delta's vulnerability, these actors framed lotus farming and the lotus landscape as a nature-based alternative to rice intensification with lotus also recognized as providing valuable ecosystem services and benefits—in Dong Thap Province and beyond, at the delta level. Within that frame, this group recognized the potential of lotus in the MDP strategy.

### 3.3.7. Traders

Yet, lotus farming could not succeed without the emergence of a bigger market, with traders, both individuals, and firms, serving key functions. For example, some traders guaranteed that outputs would be bought at fair prices, then consumed or processed in a timely way. Our interviewees recognized traders and other lotus buyers and related businesses as a particularly vital actor in creating lotus markets. Tradespeople were also said to significantly influence farmers' decisions on lotus cultivation. However, interviewees also reported that the lotus market had become unstable of late, with fluctuating demand being too high or too low. Apparently, when output prices rose, farmers grew more lotus, leading to excess supply, accompanied by the inevitable drop in prices. The large fluctuations in prices for lotus products were a cause of extreme worry among the lotus farmers interviewed. They considered this a problem requiring immediate attention from all RSGs. A solution proposed by some, was for traders to make use of their strong links to manufacturing firms to establish local capability for processing lotus raw products. Such a development, they said, could create jobs for the community while fueling local value chain development. For example, the development of new products, such as

lotus milk, could both expand the market and increase demand. Table 1 summarizes the above-mentioned perceptions of all stakeholders (via five RSGs) with their claims.

**Table 1.** Stakeholder Perspectives on Lotus Farming as an Emerging Innovation.

| Stakeholders (Relevant Social Groups) | Perceived Function of Lotus Farming as an Innovation | Claims |
|---|---|---|
| Farmers: lotus growers and laborers | Lotus farming is a valuable opportunity to improve incomes and deal with disadvantageous soil conditions | ■ Higher market price than rice (ten times higher in 2015)<br>■ Income earned periodically, giving farmers more regular income than with rice |
| Scientists: academics, researchers, and students | Lotus farming is a means to adapt to climate change and cope with floods and unfavorable natural conditions | ■ An alternative to intensive rice cultivation, which has been shown to undermine resilience to climate change<br>■ Resolves soil problems<br>■ A good opportunity for further collaboration between stakeholders toward future sustainability |
| Development partners: international organizations and NGOs | Lotus farming is a way to cope with the concurrent challenges of climate change, erratic flood seasons, and unfavorable natural conditions | ■ Provides potentially environmentally friendly livelihoods and a pathway to climate change adaptation<br>■ Resolves soil problems<br>■ One of the best ways to achieve Sustainable Development Goals |
| Local officials: local authorities and farmer representatives | Lotus farming evolved from being a profitable activity for the community to being a symbol of the region's cultural identity and heritage; it is also a means of implementing the national policy of agricultural restructuring | ■ Creates jobs<br>■ Provides a symbol of the region's great beauty and cultural heritage |
| Traders: individual traders and companies | For individual traders, lotus farming was foremost a source of monetary profit. Companies, however, viewed it as a way to achieve greater sustainability, as well as profit. | ■ Excellent market potential, as lotus is an agricultural product of which all parts are used<br>■ Higher market price than rice |

Our discussion thus far underlines that, as an emerging technology, lotus farming underwent a continuous process of social construction and development, influenced by a wide range of key stakeholders. Eventually, lotus farming did reach a point of stabilization, albeit to a low degree; that is, no reliable lotus farming model has been achieved. In terms of the SCOT lens, lotus farming has not yet reached closure. Crucial concerns at the time of our research were the unstable market prices and the serious disease affecting the lotus rhizome. Thus, the technology design still seems to be under development.

*3.4. Stakeholders' Influence on Technology Development*

All of our interviewees considered the most influential stakeholder group to be the development partners. This development partner group, they said, was instrumental in maintaining, expanding, and promoting the lotus farming model. The farmer group framed lotus as a valued opportunity for improved incomes and for coping with unfavorable soil conditions. Scientists initially framed lotus farming as a means to adapt to climate change and cope with flood extremes. Later, with the engagement of the development partners and

local officials, the multifunctional opportunities presented by lotus farming became more widely recognized. Local officials played a role in connecting farmers to the other groups. Development partners, such as the scientists, were interested in lotus as an innovation to adapt to climate change, resolve soil problems, and deliver sustainable livelihood opportunities on the regional scale. Meanwhile, local officials supported the development partner in initiating program activities. Local officials also cast the innovation in terms of national policy and development targets. Indeed, lotus farming progressed from being viewed as a profitable activity for the community to being upheld up as a symbol of the region's cultural identity and heritage. This frame facilitated the innovation's continued development. A synchronized effort emerged between scientists and local officials to promote lotus farming as an innovation. Meanwhile, traders, acting solely in response to the commercial opportunity brought by lotus, regardless of the climate adaptation aspect, found support in the frames of the other groups. However, within the trader group, differences were found. Companies expressed more concern about sustainability and understood better how useful the technology could be, especially in the context of climate change, while individual traders defined lotus farming foremost in terms of monetary benefit.

### 3.5. Influence of Available Resources on Technology Development

Interview subjects mentioned six resources as key to the development of lotus farming: technologies/knowledge, infrastructure, natural conditions, institutional mechanisms, human labor, money, and the market. Concerning technologies/knowledge, farmers said that they mainly "learned by doing", gaining valuable experience over time, crop by crop. That experience, they said, was vital in enabling them to improve their lotus production and yield.

Infrastructure was needed as well; specifically, dikes of varying heights to control the level of water entering the lotus fields. Such dikes could be constructed as road footings, with sluice gates installed so that farmers could control the water entering and leaving their fields. However, the need for such infrastructure somewhat contradicts the recent strategy of the Vietnamese government to pursue sustainable development by preserving "natural conditions" on the delta. Natural conditions have in fact proven to be an obstacle to the expansion of lotus farming. Lotus cannot be grown on high terrain, but only in lowlands that are flooded annually. Yet, if floodwaters are too high or too low, lotus can be negatively affected. Moreover, in recent years lotus farmers have grappled with a fungal disease which they attribute to "natural conditions"—though it must be said that the disease could originate from some other source, such as seed or cultivation techniques. Although the disease has been identified and studied, and some remedies proposed by CTU scientists, the issue remained unresolved for some farmers at the time of this writing.

Institutional mechanisms, in the form of official interventions such as the financial support provided by local government, played an important role in the development process, though this was largely unrecognized by the lotus farmers we interviewed. Officials confirmed that further plans were in place to develop lotus farming. This points to a gap and disconnect in information sharing between farmers and officials. If this gap were filled, lotus farming development could likely be accelerated.

Human labor remained a crucial resource because lotus cultivation cannot proceed without a sufficient supply of labor. Presently, fewer workers are willing to do the heavy work that lotus cultivation entails, as the pay is low and "easier" jobs are often available. This problem, according to some interviewees, could be solved with an institutional mechanism to assemble dedicated labor teams with access to mechanization, such as husking machines for peeling the seeds, alongside worthwhile compensation.

Money is a resource that cannot be overlooked. Lotus farmers would like to see budgets made available to financially support their operations, to guarantee output markets and prices, and to support the labor teams mentioned above.

Finally, the market has proven to be a key factor that significantly influences lotus cultivation. The three interviewees representing the trader's group said that they had

started out as general traders, buying and selling various products to earn an income. One of these traders had previously been a lotus farmer. Upon starting trade in lotus products, they recognized the strong potential of the lotus market and sought to develop it. Their first lotus product was lotus milk. After that, they expanded into salted lotus seeds, followed much later by lotus tea and lotus milk powder. Farmers came to the traders for sales and marketing. Deals were decided by both sides' agreement on the price. Traders also bought lotus products to sell on to "bigger" traders if this could be done profitably. When asked about the stability of the lotus market, traders mentioned price fluctuations, which they said were out of their control. The market decides the price of lotus, they said. Prices were said to be dependent on the supply and demand principle; if supply was high, the price dropped. All of the traders expressed general satisfaction with their business, thanks to the high potential for profit. As one trader said, "I found no failure in doing lotus, either planting it or trading it."

Table 2 shows that the first rice crop brought greater profit than the second rice crop. Lotus farming brought higher profit than a second rice crop. Lotus prices fluctuated depending on both demand and environmental factors such as the timing of floods and the incidence of plant disease. In 2018, farmers in An Giang Province earned a good profit from their lotus crop—higher even than the first rice crop—while Dong Thap farmers lost much of their lotus crop due to serious disease.

**Table 2.** Profit from Crops in Study Area, i.e., Rice, Lotus and Other Products, surveyed on December 2018 (US$/ha).

| Crop/Product | An Giang | Dong Thap | Average |
|---|---|---|---|
| Profit from first rice crop (winter-spring) | 794.24 | 553.59 | 709.30 |
| Lotus farming profit | 911.16 | 62.08 | 495.83 |
| Fish farming profit | 168.71 | 354.16 | 175.01 |
| Profit from second rice crop (summer-autumn) | 356.34 | 260.80 | 357.05 |
| Profits from other livelihoods (duck/frog/wild fish) in flood season | 282.54 | 224.25 | 264.60 |

*3.6. Closure or Stabilization of the Technology*

Since the first farmers in this region started lotus farming, in about 1980, the farming model has been recognized as a valued innovation. Commercial lotus farming was initially pursued via the interventions of the Cao Tung Company and Sachi. Financial support, though meager, provided by the government, served to clarify the semiotic structure emerging around this farming model. It also brought stabilization of the technology's meaning. The support gave farmers confidence in lotus farming and motivated them to continue the model. Lotus farming was initially viewed as a way to earn a living in a region with unfavorable soil conditions. Soon, however, it was also valued for its adaptability to changing flood levels. Furthermore, it was found to bring higher incomes and thought to be environmentally friendly as well. At the early stage of lotus farming development, the idea of "adapting to climate change" had not yet taken off. Lotus farming merely met the needs of the community better than rice cultivation. Yet, the multifunctionality of lotus farming went on to attract an array of other RSGs, especially those concerned about the environment, sustainability, and climate adaptation. The semiotic structure of lotus farming gained stability from the financial support and verbal encouragement of local officials—themselves stimulated by the mandates of the central government via national policies and plans. The fact that lotus farming was more profitable than rice farming attracted the attention of scientists and development partners. The meaning of the technology thus coalesced and stabilized, with this stabilization also narrowing the range of RSGs that could influence the further design of the innovation.

Eventually, a project sponsored by IUCN created more opportunities for the development of lotus farming as an innovation. The project invited more farmers to join in

and introduced state-of-the-art tools such as a monitoring system to test soil and water quality. These helped to make the innovation even more successful. Many activities were undertaken, including training, conferences, workshops, group discussions, and field trips with the IUCN team and experts from CTU. The RSGs joined together in a search for ways to equip farmers with better knowledge and techniques for sustainable development of the lotus farming model. This culminated in a design of lotus farming that propelled its development on a particular path. The next steps in the process would involve bringing in new actors and groups, and initiation of the process of building a "reified", or concrete, form of the emergent innovation. Yet, [32] cautioned that "the stabilization of an artifact will always be a matter of degree" and "complete closure does not happen" [33] (pp.18). We see this in the lotus case. Though actors still differed somewhat on the innovation's meaning, all RSGs agreed that lotus farming worked for them. Closure was not achieved; however, as further design modifications were still needed, which could result in new RSGs entering the scene.

Our analysis does indicate that lotus farming has reached stabilization, albeit at a low level, as all RSGs pointed to lotus farming as offering the best potential for the future. At the same time, we found that alternatives, particularly a second rice crop, have become less important. In summary, though our RSGs framed lotus farming differently, the innovation's development brought a diminishment of interpretive flexibility, and the meaning of the innovation coalesced. Among lotus farming stakeholders, all came to accept the innovation's value. This was in no small part due to the multifunctionality the innovation served.

## 4. Discussion

### 4.1. An Emerging Technology with Development Potential

Lotus farming as innovation was locally initiated, with the starting point for development being "to resolve soil problems". Stakeholders' perceptions of the technology became more diverse with the involvement and interventions of more stakeholder groups. Unexpected "difficulties" arose, however, including the departure of the Taiwanese company, lotus disease, erratic market prices, and dwindling labor availability. These set the development process on a new path from time to time.

Farmers welcomed lotus farming as a valuable innovation for the local community. Eventually, farmers realized that lotus not only could bring improved livelihoods and better incomes, but also offered a means to adapt to climate change, to cope with floods, and to resolve unfavorable natural conditions. Lotus farming became most highly stabilized for this RSG. Nonetheless, the unexpected difficulties that occurred caused farmers to look at lotus farming differently and prompted some to switch back to rice. Farmers were particularly concerned about their ability to sustain lotus cultivation. Eventually, other groups—namely local authorities, scientists, and development partners—invested serious effort and concern. Their interventions brought renewed impetus for the continued development of lotus. As the technology was further developed, farmers branched out into a diversity of lotus farming types, including lotus-ecotourism, which presently is considered the preferred model due to the steady profit it offers. Lotus-fish was also viewed as advantageous, as the combination with fish delivers a better income than lotus alone. However, these two models were not considered "sustainable" in terms of coping with climate change. In fact, lotus farming can be viewed as a relative failure in the research area, compared with other provinces, particularly An Giang, Can Tho, Vinh Long, and Long An. Most of our interviewees mentioned the success of An Giang Province in implementing lotus, as this province was key in providing good lotus seeds for other provinces. Local people in Dong Thap Province were very positive about lotus and expressed a strong willingness to work with it. However, conditions outside their control, particularly the lotus disease and the erratic market, had left them deflated. This underlines the underdeveloped state of the technology's design. Before the intervention by the scientist group (CTU in particular), little R&D funding had been available to investigate

the lotus rhizome disease and other risks and to find solutions. Much more funding is still needed. Investment in lotus pales in comparison to rice, which has received hundreds of millions of dollars in R&D funding and assistance, with dedicated research facilities such as Cuu Long Rice Research Institute, active over many decades. Both development partners and government are active in rice R&D, seeking to anticipate and reduce risks posed by diseases, pests, environmental conditions, and other threats. With greater investment at the predevelopment stage, lotus farming could turn a corner and take off as a profitable, innovative alternative.

Lotus farming was ultimately framed as a good income-earning farming model for the local community in the research area. While the innovation proved sustainable, it remained underdeveloped at the time of our survey. Indeed, farmers were less attracted to lotus in the later study years, as other crops, such as rice, were considered better options. This was due to the fluctuating market prices and labor shortages, which remained hurdles blocking the development process. With effective support from development partners, especially international organizations, the development of lotus farming could be advanced. To facilitate the process, local authorities should take decisive action to initiate engagement and collaboration among relevant stakeholders. Especially important will be to establish a platform to foster interaction between farmers and traders with fair access and a benefit-sharing scheme. Our findings also suggest a potential for lotus value chain development on the local scale, in view of lotus' multiple advantages, particularly in relation to climate change adaptation. Further technology development can be facilitated through stakeholder dialogue geared toward the advancement and maintenance of lotus farming. Our findings furthermore support the argument that it is important to look beyond what we see in the field and explore how stakeholders interpret and negotiate various aspects of emerging technology. Lotus farming as a technological artifact, in this study, was not found to be a fixed entity yet, despite a degree of acceptance among all stakeholders. All RSGs expressed appreciation for the outcomes of the lotus farming development process, at every phase of the development trajectory. The technology has undergone modifications and alterations and has been shaped by continuous interventions. This result is, importantly, in compliance with the stated dynamic nature of the Mekong delta as a system, which is at a "constant state of evolution". As a delta, its nature is framed as an intertwined and dynamic entity, consisting of agricultural, cultural, market, and water systems that have been constantly evolved [34,35]. Therefore, this finding points to the potential value of providing more room and more opportunity for additional relevant stakeholders to step in, shape, and reshape lotus farming in a way that fits their interests. This finding substantiates the SCOT perspective. Indeed, lotus farming as an artifact cannot be perceived as a single technology that was developed on its own. Rather, it was shaped by many social factors and the interactions that took place within the social network.

### 4.2. Implications for Policy, Practice, and Methodology

Since the Vietnamese government announced its intention to promote more diversified, sustainable, and adaptive livelihood models, in accordance with the MDP, lotus farming has received significant attention and investment. Our research demonstrates that lotus farming originated in a context of poor farming and livelihood conditions, and was improved through local experience and knowledge. A key finding is that the innovation's development stabilized when other relevant social groups—in addition to local farmers—became involved. Among the main stakeholder groups, our interviewees identified the most important actors as supporters of the technology, particularly international partners, local officials, and scientists. All of these had a significant influence on farmers' decisions to take up or continue lotus farming. One important actor remained largely missing from the support network, i.e., the private sector (enterprises). Their involvement could bring about a more stable lotus market. This stakeholder group could be attracted through incentive programs, involving, for example, government financial stimulus and support.

In practice, lotus farming has undergone considerable development, but this development is presently suspended due to the many hurdles that the present stakeholders have been unable to overcome. The design, or way in which lotus farming is practiced, changed over time, evolving with circumstances and societal demands. Alongside these changes, the interests of the relevant actors changed as well. Therefore, lotus farming today is very different from lotus cultivation 30 years ago. A major recommendation for policymakers is to invest in sustainable maintenance of this farming model, while also seeking to expand involvement to more actors, for example, by providing attractive modes and incentives for cooperation. Furthermore, government policies, such as tax reductions, should be considered to fuel the creation of a local value chain, thus attracting more enterprises to the market for lotus products. To the innovation implementers, although it is considered tough to introduce "innovation", especially in a rigid, top-down planning system and political context, such as that in Vietnam, the research findings suggest a new orientation, which is a call for more proactive involvement of businesses and private sector. While international actors tried to push the Vietnamese sector toward a more ecologically sustainable orientation, the realization of such a shift requires a mediation process in which both producers and consumers are influenced by various means to stabilize lotus markets and to create local production chains.

In terms of methodology implication, this study contributes to the existing literature body in the VMD, especially for further innovation-related studies. Since a historical approach was employed, a narrative story of lotus farming is built-up and interpreted via SCOT lens. This approach has shown its novelty as helpful in looking at innovation as not a fixed artifact, but as a technological artifact embedded in a complex interaction of a social network. The existing literature concerning innovations in the Mekong delta management has been found to be very few. References [7,36] already analyzed some innovative ways of farming that have been labeled as "innovation", but these works mainly looked at innovation as "innovative practices that are associated with the improvement of rural livelihoods". It was also highlighted in these works that the embedment of innovation in policymaking. "Innovation" has been a prominent theme for some projects (i.e., [37]) without looking at its development from SCOT perspective. Many recent studies looked at "new farming" that copes with climate change negative impacts without mentioning "innovation" (i.e., two of the instances are rice shrimp [38,39] or mangrove shrimp [40]). Further application of this methodology (a historical approach embedded in SCOT lens) could be considered for these specific alternatives to rice cultivation.

## 5. Conclusions

Within our UDW project, we conducted other studies exploring the roles of innovations in the strategic delta planning framework; the present study examined whether one specific innovation, lotus farming, could be viewed as successful in terms of its acceptance by relevant stakeholders. Lotus farming emerged as an attractive nature-based alternative to rice for the research area and has since received increased attention from both international and domestic actors. The MDP and climate change adaptation discourse provided a new frame and impetus for initiatives sponsored by IUCN and the Coca-Cola Foundation, as well as local government, to support and advance lotus cultivation. Yet, the success of this farming model is not a done deal, due largely to the failure to manage external factors, such as erratic prices (which could be stabilized with appropriate market development); labor shortages (which could be solved by establishing government-sponsored dedicated labor teams); inappropriate water management schemes (which could be solved with suitable dike infrastructure); and lack of horizontal and vertical integration in the value chain.

Although lotus development has not reached closure, the model has high potential, particularly in relation to the diversity of the lotus market, as all parts of the plant can be used, and lotus fields offer high aesthetic and symbolic value. Moreover, lotus products can be labeled as "green". The meaning of lotus cultivation has coalesced over time, as the

innovation was recognized as a means to cope with the floods, to adapt to climate change, to bring in better incomes, to create jobs, to improve water quality (by washing out acid sulfate), to diversify livelihoods, to restructure the agricultural system, and to symbolize local cultural identity and heritage (lotus is Vietnam's national flower). Despite the difficulties faced by lotus farmers, the lotus farming model has served an important community need. Yet, throughout the research period the technology remained in a constant state of flux, with continued development and adaptation, resulting in the technology taking different forms, each emphasizing and adapted to different circumstances and services. The technology has not reached closure but remains continuously subjected to modifications and change to enable it to work better in the context of the VMD.

With increased investment, the quality of local lotus products could be brought up to the requirements of the international market. For example, a training or lotus farming manual might provide some help. Other aspects that should be considered are effective linkages between farmers and traders, particularly the companies that process raw lotus products (now mostly located in Ho Chi Minh City and Binh Duong Province). Farmers would benefit from direct connections to these large buyers to market their products. Furthermore, the idea of a government-sponsored labor team was raised, to help solve the problem of labor shortages.

The current study found a range of social actors that supported lotus farming's development. We can conclude that lotus farming has experienced continuities and discontinuities in technology development, but nonetheless has great development potential. This finding is of wider importance because, as delta researchers, we propose innovative ideas not only to be written in policy plans but, more importantly, for implementation to achieve sustainable deltas worldwide. Our findings in this study echo the social construction of technology perspective, as we found lotus farming to be a technological artifact that did not develop on its own. Rather, its development was driven by stakeholders' interpretations and the meanings they assigned to it. In this sense, it has been socially shaped and continues to be reshaped. This conclusion is an important one for a strategic delta planning process, as it confirms that an innovation takes time to travel from formulation to the field. Moreover, no innovation can be implemented without acceptance by the relevant stakeholders. As of this writing, many aspects of the lotus farming model remained unclear or underdeveloped as its design still allowed many windows for actors to insert their ideas for further development. This points to the important role of stakeholders' perceptions (especially farmers) in creating satisfying outcomes for an innovation and enabling multiple co-evolution of a technology. The regional planning process recognizes lotus as a highly adaptive, innovative technology. To achieve this potential, greater government support is needed, to attract more actors to the development of this innovation. Increased research attention is also needed to identify the right models to make this technology work.

**Author Contributions:** Conceptualization, H.T.M.V., G.v.H., P.H.; methodology, H.T.M.V., G.v.H.; validation, H.T.M.V., G.v.H., P.H., A.W., Q.H.N.; formal analysis, H.T.M.V.; investigation, H.T.M.V., A.W.; resources, H.T.M.V., A.W.; data curation, H.T.M.V.; writing—original draft preparation, H.T.M.V.; writing—review and editing, H.T.M.V., G.v.H., P.H., A.W., Q.H.N.; visualization, H.T.M.V.; supervision, G.v.H., P.H.; project administration, G.v.H., P.H.; funding acquisition, G.v.H., P.H. All authors have read and agreed to the published version of the manuscript.

**Funding:** This research was funded by the Urbanising Deltas of the World Programme of the Netherlands Organisation for Scientific Research (NWO), project number W 07.69.106.

**Acknowledgments:** Our special gratitude goes to the interviewees for sharing their insights on the process of lotus farming innovation. Special thanks go to IUCN Vietnam for on-the-ground support for the interviews and field trips. The last author is supported by the DUPC2 project "Flood-based farming systems for enhancing livelihood resilience in the floodplain of upper Mekong delta". We also thank the two anonymous reviewers for their constructive comments on this manuscript.

**Conflicts of Interest:** One of the co-authors, Andrew Wyatt, has affiliations with the IUCN, an organization that promoted lotus production and received funding from Coca-Cola Foundation. However,

all opinions presented in this manuscript belong to the authors alone, and not any institution to which they are or were affiliated. So the authors declare that they have no competing interests. Additionally, the funders—NWO had no role in the design of the study; in the collection, analyses, or interpretation of data; in the writing of the manuscript, or in the decision to publish the results.

## Appendix A. Interview Guide

Main questions (flexible, to be modified according to the interviewee type).

Topic 1. Soft implementation: how the lotus farming model affects people's thinking in improving local livelihoods to cope with floods and directions to sustainability?

- Are you a lotus farmer? What type of lotus farming you are doing and why not others?
- For how long have you done this? How did you start with lotus, who introduced it to you?
- What are the advantages/disadvantages of farming lotus?
- What is the current status of practicing this model at your locality?
- What changes in the lotus farming model will occur in the future when compared to the current farming models?
- Who is knowledgeable about lotus farming? What kinds of people are changing their understanding of dealing with floods and climate change?
- Who is supporting the practice of this model? Who does not? And who is changing?

Topic 2. Hard implementation: How is the lotus farming model implemented in practice?

- Which (formal) decisions and activities include adaptive agricultural models in general and flood-based farming systems in particular? (e.g., plans, policies, regulations, research programs)
- Where is the lotus cultivation model implemented in the Mekong Delta? What are the criteria for applying lotus cultivation models in these areas?
- What technologies were used when implementing the lotus farming model?
- What creative ideas are relevant when implementing the lotus farming model?
- What are the recommendations for applying and replicating the lotus cultivation model to other areas?

Topic 3. Historical perspective

- When did you know about lotus farming?
- As you know who started the lotus here, what was the reason they planted lotus at that time? What did they do before planting lotus?

Topic 4. Network

- In practicing these models, who are you collaborating with?
- Do you find yourself collaborating with / or working with someone else to improve the lotus farming development?
- Specifically, among the social groups outlined below, which are the relevant groups that may affect the development of lotus farming?
  - Group of farmers
  - Local government groups
  - Researchers/scientists group
  - Business group (could be a company or a buyer)
  - Group of foreign organizations
  - Non-governmental organization
- (Is there any other relevant group, according to you?)
- Which of the five groups according to you think the most important role to develop lotus cultivation, which group contributes the least?

Topic 5. Future potential

- Do you continue to farm lotus in the future, why?
- If you do not continue with lotus, what are your alternatives?
- Other things to mention
- Do you have any specific aspirations or suggestions for improved lotus cultivation and increased productivity?
- Do you have any specific aspirations or suggestions to promote the construction of a lotus product purchasing and processing company on the spot?
- Assuming you do not have the expected returns from practicing the model, will you continue it? Why so?

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
