# Peer review of "The Emergence of Lotus Farming as an Innovation for Adapting to Climate Change in the Upper Vietnamese Mekong Delta"

_land, doi:10.3390/land10040350_

Round 1

Reviewer 1 Report

General comments:

The manuscript, entitled " The Emergence of Lotus Farming as an Innovation for Adapting to Climate Change in the Upper Vietnamese Mekong Delta " focuses on local perceptions of such a system in Dong Thap Province, a lowland, flood-affected area of the upper VMD. Specifically, we explored lotus farming as a potential flood-based adaptive livelihood model for the region. There are some issues with this manuscript, mainly related to the readability and composition of the manuscript.

 Please review the quality of your English throughout the manuscript.

I would like to see some concluding remarks and recommendations in the abstract section.

Specific comments:

Point 1: I recommend to the authors in the introduction part to add more literature review in order to emphasize the need and originality of your research.

Point 2: Rephrase sentences indicted in Lines 113-116.

Point 3: I recommend to the authors the research design be mixed type (that is, quantitative and qualitative). But this study focused only qualitative part that lacks the quantitative aspect.

Point 4: Discussion section (Line 573), this is an important part of the study and I suggest the authors compare your results with other similar studies in order to emphasize the originality and novelty of the study.  

Point 5: Line 674 no need for citations in the conclusion section.

Point 6: I suggest write some recommendations to the concerned policymakers and implementers.

Author Response

The Emergence of Lotus Farming as an Innovation for Adapting to Climate Change in the Upper Vietnamese Mekong Delta

Response to #reviewer 1 comments

Dear reviewer 1,

On behalf of all co-authors,

Firstly, we would like to thank you very much for your constructive feedback on our manuscript in which we analyze local perceptions of lotus-farming as a potential flood-based adaptive livelihood model for Dong Thap province (a low land and flood-affected area of the upper Vietnamese Mekong Delta (VMD)). We appreciate all of the feedback, comments, and suggestions that have helped to improve our manuscript significantly, especially regarding the readability and the composition of the manuscript. We are very grateful for the time you spent reading and commenting on our manuscript. We also reflect this gratitude in the rewritten Acknowledgements.

Secondly, below we present the changes in the manuscript, responding to your comments and explain how we address these points.

Point 1: I recommend to the authors in the introduction part to add more literature review in order to emphasize the need and originality of your research.

We very much appreciate the comment about adding more literature review in the introduction, to emphasize the need and originality of the research. We are now adding some sentences justifying the need and originality of our research, these can be found in lines 86-91, page 2.

Point 2: Rephrase sentences indicted in Lines 113-116.

We appreciate the suggestion of rephrasing the sentence of lines 113-116, and we did now for lines 120-123, page 3.

Point 3: I recommend to the authors the research design be mixed type (that is, quantitative and qualitative). But this study focused only qualitative part that lacks the quantitative aspect.

We understand the importance of a mixed-type of research design, that makes a stronger and more solid methodology to address the research question, however, as our study is more on the perception of grassroots stakeholders, towards the emergence and the development pathways of (one of) their daily practices, that they have not put attention on, but now being emerged and evolved and received new interests from other stakeholders. We considered thoroughly and decided to focus only on qualitative aspects. However, quantitative research could be a sound call for further research as it can confirm our findings (qualitatively) in this current study. So we will seriously take your suggestion into account, for further work with our profound thankfulness.

Point 4: Discussion section (Line 573), this is an important part of the study and I suggest the authors compare your results with other similar studies in order to emphasize the originality and novelty of the study.  

We appreciate the suggestion of making a comparison of our results with other similar studies in order to emphasize the originality and novelty of the study, and we did. This can be found in lines 686-698, page 15

Point 5: Line 674 no need for citations in the conclusion section.

We appreciate the suggestion of removing the citations for line 674, and we did, now in line 701, page 16

Point 6: I suggest write some recommendations to the concerned policymakers and implementers.

Yes, this is a very good point, that we are grateful to, and now in the revision, we added some recommendations for the concerned policy makers (already existed) and implementers. This can be found in lines 675-682, page 15

General comment on English editing and the Abstract:

Thank you very much for the suggestion, we have carefully checked again the English, and some corrections and rephrasing have been made. We now also added some concluding remarks into the Abstract (lines 27-33) page 1

Thank you!

Reviewer 2 Report

This is useful paper worthy of publication.  My only concern relates to potential or actual conflicts of interest given one of the authors, Andrew Wyatt is employed by IUCN, an organisation that promoted lotus production and received funding from CocaCola. While this made clear in the text,  this should be made clear, in the publication section on conflicts of interest as it appears to a be a direct conflict.

My other criticism is that paper lacks a clear framing about the dynamic nature of agricultural, cultural, market and water systems. These are in a constant state of evolution. I suggest minor revision to bring this dynamic perspective to way the evolution of farming in the delta (and elsewhere) is evolving.

If the paper is to be revised it would useful to refer to D. Biggs, F Miller, F Molle, HOANH, 

"The delta machine : water management in the Vietnamese Mekong Delta in historical and contemporary perspectives"

 (2009) In book: Contested Waterscapes in the Mekong Region: Hydropower, Livelihoods and Governance

This kind of analysi of the Mekong delta could provide useful background about the nature of the dynamic state of the Delta - the machine that never stops. 

Another useful reference on Delta, agriculture people interactions is  Biggs, D. 2014 Promiscuous Transmission and Encapsulated Knowledge: A Material-Semiotic Approach to Modern Rice in the Mekong Delta In Rice: Global Networks and New Histories, (Eds), F. Bray, P. Coclanis, E. Fields-Black & D. Schäfer (pp. 118–137). Cambridge, UK: Cambridge University Press.

Author Response

The Emergence of Lotus Farming as an Innovation for Adapting to Climate Change in the Upper Vietnamese Mekong Delta

Response to #reviewer 2 comments

Dear reviewer 2,

On behalf of all co-authors,

Firstly, we would like to thank you very much for your constructive feedback on our manuscript in which we analyze local perceptions of lotus-farming as a potential flood-based adaptive livelihood model for Dong Thap province (a low land and flood-affected area of the upper Vietnamese Mekong Delta (VMD)). We appreciate all of the feedback, comments, and suggestions that have helped to improve our manuscript significantly, especially the framing about the dynamic nature of the Mekong delta system, literature, and the conflicts of interests. We are very grateful for the time you spent reading and commenting on our manuscript. We also reflect this gratitude in the rewritten Acknowledgements.

Secondly, below we present the changes in the manuscript, responding to your comments and explain how we address these points.

First of all, we are very thankful for your first concern, regarding the potential or actual conflict of one of the authors, Andrew Wyatt who has affiliations with the IUCN, an organization that promoted lotus production and received funding from Coca-Cola Foundation. However, all opinions presented in this manuscript belong to the authors alone, and not any institution to which they are or were affiliated. So the authors declare that they have no competing interests. Additionally, the funders - NWO had no role in the design of the study; in the collection, analyses, or interpretation of data; in the writing of the manuscript, or in the decision to publish the results.

We appreciate very much your second constructive remarks, which significantly help us to enrich the discussion section, while we talked about the evolution between (technological) innovations and the strategic delta planning processes. We agree on the “constant state of evolution” of the dynamic nature of the Mekong Delta system as a whole entity, thus taking a perspective on framing such a dynamic nature, to reflect the evolution of farming in a delta, is very helpful.

So we add some sentences about this dynamic perspective in the discussion section, using your recommendation on the works of Biggs (2009 and 2014). This can be found in lines 640-643, page 14. In fact, these works played crucial roles in the literature part of other publications of our team:

  • Hoang Thi Minh Vo, Gerardo van Halsema, Chris Seijger, Nhan Kieu Dang, Art Dewulf, Petra Hellegers, 2019, Political agenda setting for strategic delta planning in the Mekong Delta – Converging or diverging agendas of policy actors? Journal of Environmental Planning and Management https://doi.org/10.1080/09640568.2019.1571328
  • Chris Seijger, Hoang Thi Minh Vo, Gerardo van Halsema, Wim Douven, Andrew Wyatt, 2017, Do strategic delta plans get implemented? The case of the Mekong Delta Plan, Journal of Regional Environmental Change https://doi.org/10.1007/s10113-019-01464-0

And also, in the Introduction part of the 1st author’s Ph.D. dissertation.

Thank you!
